# Corporate Editors in the Evolving Landscape of OpenStreetMap

**Jennings Anderson** [1,*] , **Dipto Sarkar** [2] **and Leysia Palen** [1,3]

1    Department of Computer Science, University of Colorado Boulder, Boulder, CO 80309, USA;
     leysia.palen@colorado.edu
2    Department of Geography, National University of Singapore, Singapore 119077, Singapore;
     dipto.sarkar@nus.edu.sg
3    Department of Information Science, University of Colorado Boulder, Boulder, CO 80309, USA
*    Correspondence: jennings.anderson@colorado.edu

**Abstract:** OpenStreetMap (OSM), the largest Volunteered Geographic Information project in the world, is characterized both by its map as well as the active community of the millions of mappers who produce it. The discourse about participation in the OSM community largely focuses on the motivations for why members contribute map data and the resulting data quality. Recently, large corporations including Apple, Microsoft, and Facebook have been hiring editors to contribute to the OSM database. In this article, we explore the influence these *corporate editors* are having on the map by first considering the history of corporate involvement in the community and then analyzing historical quarterly-snapshot OSM-QA-Tiles to show where and what these corporate editors are mapping. Cumulatively, millions of corporate edits have a global footprint, but corporations vary in geographic reach, edit types, and quantity. While corporations currently have a major impact on road networks, non-corporate mappers edit more buildings and points-of-interest: representing the majority of all edits, on average. Since corporate editing represents the latest stage in the evolution of corporate involvement, we raise questions about how the OSM community—and researchers—might proceed as corporate editing grows and evolves as a mechanism for expanding the map for multiple uses.

**Keywords:** OpenStreetMap; corporations; geospatial data; open data; Volunteered Geographic Information

## 1. Introduction

OpenStreetMap (OSM) is a freely available and openly editable map of the world founded in 2004 by Steve Coast in response to the prohibitively expensive geographic data owned by the Ordnance Survey [1]. Since this time, OSM has grown into the world's largest Volunteered Geographic Information (VGI) platform. OSM is comprised of the consumable product—the mapped, geographic data produced by millions of people around the world—and the massive community that maintains it. At its technical core, OSM is a geospatial database with billions of entries that denote hundreds of millions of physical objects in the real world. Several researchers have commented on the growth in the volume and the evolution of this geographic content in terms of accuracy and completeness [2–6]. The constantly-evolving map is supported by a growing community of mappers with a variety of motivations [7]. In addition to individual mappers, various groups formed around OSM also provide clues about the diversity of interests in the OSM community. These include for-profit organizations that use the map-data, organizations such as the Humanitarian OpenStreetMap Team (HOT), which creates geospatial data both in preparation of and response to humanitarian crises around the world, or the

many formal and informal local OSM communities that organize *mapping parties* and other events to encourage participation and data contribution. As such, OSM can be described as a "community of communities" that curate and edit map data on a single platform, compelled by a range of individual and shared motivations, but with the over-arching objective of creating a freely accessible, open, and editable map of the world [8]. The continued growth of OSM is a testament to the idea that maps are never fully formed, and are thus an ever-evolving product of embodied, social, and technical processes [9]. Maps represent snapshots of the moment, reflecting the values and priorities of their creators. The various communities within OSM edit the map with different goals and motivations with the hope that the common platform results in a uniform product useful for all. The ongoing efforts of this "community of communities" make OSM a constantly evolving map-of-the-moment adapted to the requirements of the day.

The last two years have seen major growth of a particular type of community: corporate editors. These are paid editors that curate the map professionally. While numerous for-profit corporations have always been involved in OSM—typically through using OSM data in their services and products—the rapidly increasing number of paid-editors on the platform is new and has become a contentious issue for some in the community. Presumably, the corporations employing these editors are investing in OSM in relation to their product. For example, some core Mapbox products rely on maps built on OSM data. As such they were one of the first companies to engage in this activity, beginning as early as 2014. Other companies, such as Amazon Logistics, claim to use some OSM data in their internal routing algorithms. In turn, they contribute back information from their drivers to improve the vehicle routing abilities of OSM data [10]. In this article, we identify ten corporations that transparently employ teams of professional editors. We explore the editing activity of each team to better understand the impact on the map and community. Though some editing mishaps have made the OSM community suspicious of corporate editing, guidelines around transparency and community engagement are now in place that these corporations attend to—and in so doing, make the usernames of their editors available. To the best of our knowledge, this is the first article exploring the role and contributions of corporate entities editing OSM at scale. We consider the discourse about corporate involvement in OSM to inform and contextualize quantitative analyses of the OSM database to measure the global footprint of the ten companies.

## 1.1. OSM Contributors

OSM relies on volunteer contributions to build and curate the map: specifically, this means that OSM does not offer financial incentives to mappers. Currently, there are more than 5 million registered users, over 1 million of whom have edited the map. The growth of the entire OSM community is shown in Figure 1. Researchers have noted the motivations for contribution to OSM as ranging from altruistic to vandalistic as a result of intrinsic self-motivations and external societal, economic, or political drivers [10–13]. The legal entity behind the OpenStreetMap project is the OSM Foundation (OSMF). OSMF is a U.K.-registered non-profit that supports OSM by fund-raising, managing servers, organizing and sponsoring conferences, and supporting working groups that attend to various business functions such as licensing, operations, or communications. OSMF is run by a board which is elected by due-paying members [14]. Membership with the OSMF is separate from having a user account on openstreetmap.org, which is required for mapping. There is no requirement to join the OSMF to be part of the OSM community (that is, as a mapper, data consumer, etc.). Though there may be overlap in personnel, projects, and donors, but there are no formal governing links between OSM subcommunities—such as HOT, local OSM groups, or companies—and the OSMF.

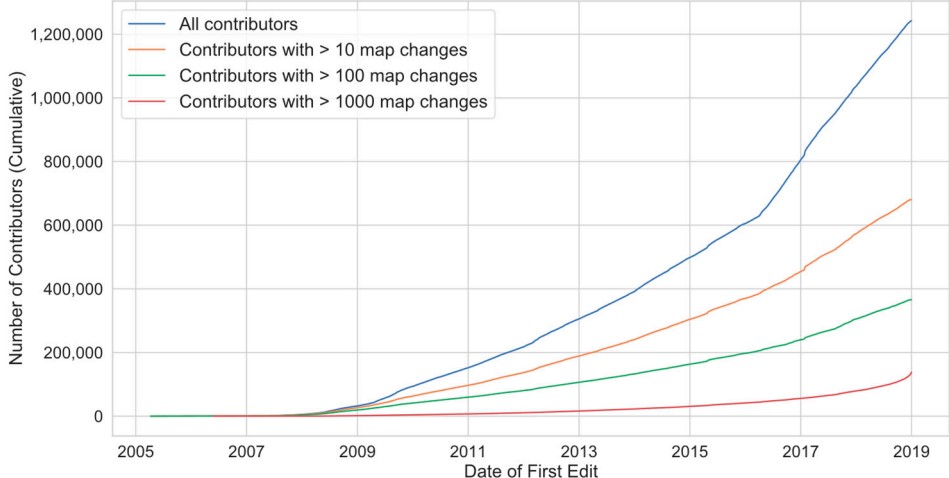

**Figure 1.** OpenStreetMap Contributors: Over 1 million users have made at least 1 change to the map. Far fewer contributors have contributed more than 10, 100, or 1000 times. Results calculated from an OSM changeset database, created from the OSM changeset files by the open source tool: github.com/toebee/changesetmd.

The response of the OSM community has been notable in the wake humanitarian crises [15,16]. In particular, HOT mobilizes and coordinates global mapping events in response to disasters, including Typhoon Yolanda (2013), The Ebola Crisis (2014), and the Nepal Earthquake (2015), to name just a few. Additionally, local OSM communities organize *mapping parties* to recruit and support new participants as well as to map previously unmapped areas [17,18]. Regional and global *State of the Map* conferences are also organized by active OSM groups, typically with support from the OSMF and regional OSM organizations. In addition to the map itself, there are active mailing lists and a wiki which also serve as venues for user contributions and discussion.

Not all users contribute equally to the map. OSM is no exception to the 90-9-1 rule found in online communities where only a small number of active contributors account for most of the contributions [19]. By our calculations for OSM, the top 1.4% of editors are responsible for 90% of all the map changes (Figure 2). On a monthly basis, approximately 1 to 13 percent of users actively contribute data [20]. Figure 1 shows that though over 1 million contributors have edited the map, less than 700,000 have made more than ten changes to the map.

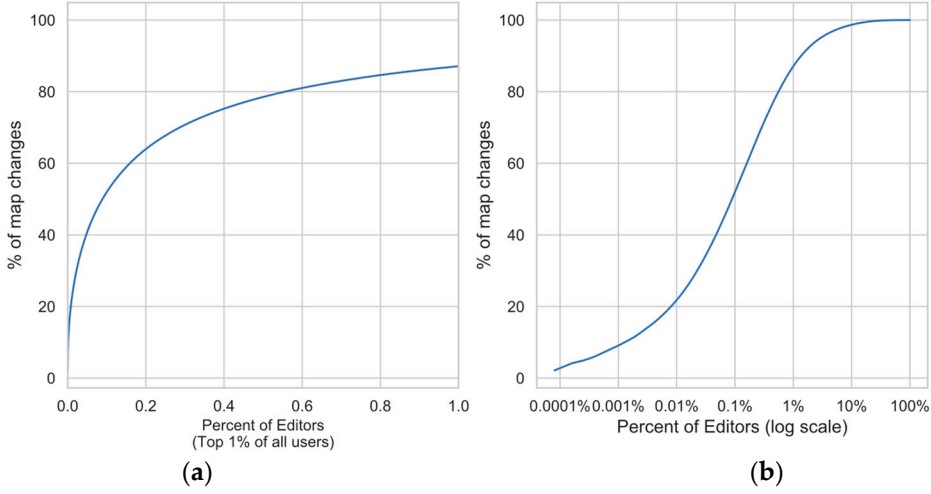

**Figure 2.** (**a**) The top 1% of users are responsible for 87% of all the changes to the map; (**b**) OSM adheres to the 1% rule: a very small percentage of the editing community contributes the majority of the data. Results calculated from the OSM changeset database described in Figure 1.

Like other online platforms, OSM also reproduces offline inequalities. Several groups of people are underrepresented, including women, people in the Global South, people of color, and non-urbanites [21–26]. The skewed participation in OSM produces several artifacts in the data [27–29]. For example, the predominance of male participation in OSM has created an apparent over-representation of features that are correlated to male interests [21]. Availability and access to the internet, technical knowledge, barriers created by the gatekeepers of the platforms, and lack of free time and opportunity to contribute have been recognized as some of the hindrances to equal participation [21,25,29–31]. In addition to systemic barriers, researchers have also highlighted that the global political landscape has significant impact on contributors and consequently, on the data produced [31–36].

*1.2. Landmark Corporate and Government Contributions to OSM*

While the rise of corporate editing teams is a new phenomenon in OSM, corporate presence is not new to OSM. For over a decade, corporations, governments, and other organizations have been heavily involved in shaping OSM as it exists today. These involvements are documented through the OSM wiki, mailing lists, and blogs, and cannot be traced through the scientific literature alone. As one example of this, the OSM founder, Steve Coast, also founded Cloudmade, a company that provided geo-services based on OSM data [37]. In this we see that special-interest groups are not new to the OSM community; corporate editorship is not simply a case of capitalist appropriation of an open data project, but rather the latest stage in an evolving project comprised of a wide-array of stakeholders, each coming from a different value system.

Next, we highlight a few key involvements of external groups that have had significant impact on shaping the community and the map since its inception. First, the ability to trace features from Yahoo! aerial images as of December 2006 removed the barrier of requiring GPS devices for contributing to OSM [38]. This enabled "armchair mappers" to create and edit data for remote locations. However, armchair mapping comes with its own set of challenges caused by georeferencing errors and temporality issues. These issues prompted OSM to come up with guidelines for tracing features [39]. Over the years, various custodians of aerial and satellite imagery—including Bing, Esri, Digital Globe, and Mapbox—have made their data available for tracing in OSM. A comprehensive list of imagery providers is maintained on the OSM Wiki [40]. Making satellite images available post-disaster has been critical in the usability of OSM for disaster response [41]. This has particularly aided the OSM community in quickly creating data for areas that lack good geospatial data during times of need [16]. Projects such as HOT and Missing Maps leverage the image tracing function to mobilize armchair mappers to contribute data for vulnerable places that lack geospatial data

Second, large data contributions have significantly increased the map data available and overall map usability. A landmark contribution of government data to OSM was the uploading of the Topologically Integrated Geographic Encoding and Referencing (TIGER) dataset produced by the U.S. Census Bureau starting in September 2007. The Automotive Navigation Data (AND) was also uploaded at a similar time, adding the road network for the Netherlands along with parts of India and China [38]. Several organizations, groups, and individuals have since contributed to OSM through large data imports. Such imports of bulk data are valuable for increasing the data volume, though integrating them with existing OSM data is challenging. For example, after the TIGER import, several compatibility errors were noted because the TIGER dataset and OSM do not follow the same road classification [42]. For managing the challenges of data integration, the community has come up with guidelines for importing government data [43]. The OSM wiki maintains a list of 'large-scale' data imports and potential data sources for import and use [44,45].

Several governments are both using and contributing to OSM. The World Bank has supported development of OSM data for both humanitarian crisis purposes and also as an ongoing effort for places that lack capabilities to develop geospatial data [46]. Government entities including the City of New York and Portland's Traffic Authority have dedicated teams responsible for improving OSM data

in their jurisdictions [47]. Previous research has described government contributions and usage of OSM data in greater detail [46–48]. Corporate entities such as Mapbox, Stamen, and Geofabrik also use OSM data and make active contribution to the database and community through various services they provide [47]. Corporate contributions to OSM data in small cities that lack good geospatial data has also been noted [26].

Even though focused attention on corporate editing by the OSM community is reaching new, visible heights, the OSM contributor network has been historically comprised of public and private entities that have participated for various reasons in a shared vision of an open map of the world. This report therefore focuses on the apparent growth of corporate involvement in the past few years, and why their growing participation through map editing may be fraught, and what this might mean for the future of OSM.

## 2. Materials and Methods

The companies examined in this report were identified through either their longtime involvement in the OSM community, noted by their continued sponsorship of the Foundation and/or conferences, or their current transparency in publicly revealing their involvement in editing the map. This comprehensive sample was made by those with the most editing activity (Apple, Mapbox, and Kaart) along with seven other corporations that the authors were able to identify through their conference participation and their publicly visible list of paid editors. In total, we identified 954 usernames associated with corporate editing. At the time of writing, we are unaware of other corporations with as much editing activity as those identified here. It is possible that there are other companies employing teams of editors, but have yet to disclose this information.

We used two types of data sources to then further examine the role of corporate editorship: public articles and data about corporate involvement in OSM, and the geospatial data created by corporate editors.

For the first source, we identify information across websites and media outlets to help trace the interest expressed by corporate editors for using and editing OSM. This information links also to publicly-available data that lists usernames of editors associated with each corporate team. It also lends insight into the motivations, the nature of edits, and the mode of edits because these companies both list and discuss specific mapping projects and their progress. The OSM sponsors list was used as the starting point for assembling a list of companies interested in OSM. Media articles were obtained when developments regarding this new phenomenon of corporate editing occurred. The authors' long-time experience in the OSM community, including personal observations at State of the Map conferences, informed the formation of the questions and interpretations.

For the second source, we use historical quarterly-snapshot OSM-QA-Tiles for quantifying where and what the corporate editors are editing on the map. OSM-QA-Tiles are vector tiles containing object level editing behavior for the vast majority of OSM data: roads, buildings, points-of-interest, etc. in an efficient, accessible form. For example, a recently modified building will exist in an OSM-QA-Tile as a polygon object with metadata including the name of the mapper that most recently edited it, the timestamp of this edit, and the current version number of the building: denoting whether this user created the building (version = 1) or edited an existing object (version > 1). We find this to be more accessible than the standard OSM data-model which requires first reconstructing the building by identifying the individual nodes associated with the object. However, an analytical weakness of the standard OSM-QA-Tiles is that map objects are unique (one version of each object), so other than knowing their current version number, objects are unaware of their own editing histories. Thus, these tilesets can only represent a snapshot in time: the most recent version of the map data. For this historical analysis, we used the historical quarterly-snapshot OSM-QA-Tiles. These tiles represent the map at the end of every quarter since 2005. Historical OSM data analysis is possible by iterating through these tiles to get quarterly development of the map. For example, if a road was created in January, and then subsequently edited three more times in April, August, and December of that year,

then each of the quarterly snapshots will include this road along with the metadata corresponding to each of these edits (e.g., usernames and date of each of these four changes). An annual snapshot, in contrast, would only include metadata for the latest edit occurring in December. Objects are edited at all frequencies, but quarterly snapshots give a finer resolution of the evolution of the map while still making global-scale analysis computationally efficient. We use the open-source Javascript framework *tile-reduce* (github.com/Mapbox/tile-reduce) to efficiently process these historical vector tilesets, following the same methodology as previous work by Anderson et al. [15].

Thus, the initial analysis of media articles, blog posts, and wiki pages enables us to position corporate editors in the context of the larger OSM community, while the evaluation using OSM-QA-Tiles quantifies the impacts to the map. To label edits as corporate, we match the usernames associated with edits with the publicly disclosed lists of usernames associated with each company. In the event a mapper edited before and/or after being employed by a company, we filter by time to count only the edits that occurred during the mapper's employment on a corporate data team.

## 3. Results

### 3.1. Observational Analysis of Corporate Involvement

We focused on the ten corporate entities that have shown significant interest in editing OSM. We highlight the announcements and coverage of this phenomenon in different news media in the past two years, then we discuss the visibility and contributions of these companies in the OSM community. We then examine the traces of these companies in the OSM data itself. Table 1 highlights how the quantity and variety of contributions from each varies dramatically.

**Table 1.** Known Corporate editing teams active in OSM. The column *OSM Foundation Engagements* shows the current affiliation with OSM Foundation and their sponsorship of State of the Map conferences. Data for edits as of January 2019 (since 2014), rounded to the nearest thousand.

| Corporation | OSM Foundation Engagement | Team URL | Team Size | Number of Edits | km of Roads Edited | Buildings Edited |
|---|---|---|---|---|---|---|
| Amazon | • Gold Corporate Sponsor (Amazon Web Services)- State of the Map 2013 | wiki.openstreetmap.org/ wiki/Amazon_Logistics | 110 | 388,000 | 120,000 | 1000 |
| Apple | | github.com/osmlab/ appledata/wiki/Data-Team | 342 | 3,944,000 | 1,643,000 | 1,156,000 |
| Development Seed | | wiki.openstreetmap.org/ wiki/DevSeed-Data | 8 | 488,000 | 62,000 | 269,000 |
| Facebook | • Gold Corporate Member Gold Sponsor-State of the Map 2018 • Silver Sponsor-State of the Map 2017 • Bronze Sponsor-State of the Map 2016 | wiki.openstreetmap.org/ wiki/AI-Assisted_Road_ Tracing | 87 | 1,106,000 | 821,000 | 1000 |
| Grab | • Gold Corporate Member | github.com/GRABOSM/ Grab-Data | 124 | 1,593,000 | 300,000 | 63,000 |
| Kaart | • Bronze Corporate Member • Bronze Sponsor- State of the Map 2018 | wiki.openstreetmap.org/ wiki/Kaart#Kaart_Data_ Team | 93 | 2,887,000 | 484,000 | 702,000 |
| Mapbox | • Gold Corporate Member • Gold Sponsor- State of the Map 2018, 2017, 2016, 2014, 2013 • Silver Sponsor- State of the Map 2012 | wiki.openstreetmap.org/ wiki/Mapbox#Mapbox_ Data_Team | 40 | 4,483,000 | 1,694,000 | 1,088,000 |

**Table 1.** *Cont.*

| Corporation | OSM Foundation Engagement | Team URL | Team Size | Number of Edits | km of Roads Edited | Buildings Edited |
|---|---|---|---|---|---|---|
| Microsoft (Bing) | • Gold Corporate Member<br>• Gold Sponsor- State of the Map 2018, 2017<br>• Platinum Sponsor- State of the Map 2011, 2010 | github.com/Microsoft/Open-Maps/wiki/Open-Maps-Team-at-Microsoft | 29 | 643,000 | 458,000 | 52,000 |
| Telenav | • Silver Sponsor- State of the Map 2018, 2017, 2016<br>• Platinum Sponsor- State of the Map 2012 | wiki.openstreetmap.org/wiki/Telenav#Telenav_folks_on_OSM | 30 | 963,000 | 336,000 | 5000 |
| Uber | | github.com/Uber-OSM/DataTeam | 91 | 464,000 | 32,000 | 349,000 |

### 3.1.1. Tracing Corporate Interest Through Media

Bing, a subsidiary of Microsoft, has contributed 125 million building footprints in the U.S. to OSM, which they extracted from aerial imagery through deep learning algorithms [49]. In addition to contributing automatically extracted and generated data, Microsoft has also assembled a team of editors to contribute to OSM. The aim of their Open Maps Team is to work closely with the OSM community to improve data quality in places of strategic importance to Microsoft [50]. The team coordinates their activities through GitHub where their team members and projects are listed. Each project is thoroughly described there. In addition, GitHub issues-tracker offers a place for community feedback and questions, which supports transparent, documented issue resolution for each mapping project (github.com/Microsoft/Open-Maps/issues). Mapbox/DevSeed, Apple, Kaart, Telenav, & Grab also use GitHub in the same way to track their projects and answer community questions. Microsoft's commitment to OSM is an extension of their support of open source projects [51,52]

Facebook's OSM contributions to date have mostly been through supervised automated contributions. They use machine learning to detect road networks from satellite imagery which are then validated and reviewed by their OSM editors who work closely with the local OSM communities. All machine-identified roads are reviewed by a human editor before being imported into OSM. Their efforts were initially focused on mapping Thailand; they have completed editing road data for all 79 provinces, adding a total of 515,306 km of road to the map [53]. They have used similar infrastructure in collaboration with HOT to contribute in the aftermath of the 2018 floods in Kerala, India [54].

One of the most valuable aspects of digital maps are navigation capabilities. However, ensuring topological and semantic rules is tedious [2,55–57]. Government and corporate data contributions, coupled with the efforts of the community to clean and integrate these data into OSM, have ensured that road network data in many places in Europe and North America are suitable for navigation. However, this is not the case for OSM data in Asia. Many Asian countries have emerged as big markets for ride sharing services such as Uber and Grab [58]. Uber has announced that it wants to migrate its mapping service to OSM; New Delhi, India will be the first city where this OSM-based service will be rolled out [59]. Uber also announced through a community posting in an OSM forum that they will involve a team of editors to improve map data specifically for navigation by modifying and adding turn restrictions, directionality, and road geometry [60]. Grab has dedicated considerable amount of effort into improving OSM data for Southeast Asia. In addition to having a team of editors, Grab has organized several mapathons in many countries for wider community engagement [61]. They have also partnered with HOT for the mapathons to ensure their edits are relevant in crisis situations [62].

Until 2018, the Mapbox data team was the most active team of corporate editors in the OSM community. Mapbox was one of the first companies to employ a team of OSM-specific editors, starting as early as 2014. In late 2017, a large part of the Mapbox data-team merged with the Development Seed data team, creating DevSeed Data [63]. Like Facebook, this team is also heavily invested in machine-assisted mapping: using machine learning to help their data team identify features to map.

Kaart drives vehicles all over the world to capture road networks and ground-level imagery to improve OSM [64].

The OSM community has been divided about its policies to enforce transparency and accountability for what they refer to as "organized editing," which captures mapping activities by both nonprofit (e.g., HOT humanitarian mapping) and for-profit groups (i.e., the groups described here). In a 2017 survey by the OSM Foundation, 43% of paid editors—compared to 17% of all respondents—opposed having policies that guide editing activities [65]. Ultimately the OSM Foundation produced the *Organized Editing Guidelines* in November 2018, the goal of which is to ensure meaningful, transparent participation from large editing teams [66].

Though these ten corporations have been transparent about their editing activities of OSM, there have been mishaps regarding editing conflicts with the community. For example, Grab was in the spotlight in late 2018 for the oversight by their outsourced editors for overriding volunteer contributions with incorrect edits in Thailand. This incident brought unresolved attention about why companies (like Grab) which do not seem to be using OSM in their product are interested in contributing and improving OSM. One Bangkok-based OSM enthusiast speculated that Grab (and Uber) were using OSM data for improved routing in their applications without attribution [67,68].

### 3.1.2. Contributions to the Larger Ecosystem and Community Participation

The involvement of these corporations in OSM extends beyond editing the map. Many of the open source software tools in the OSM ecosystem are developed and maintained by their employees. For example, iD—the user-friendly, in-browser editor incorporated into openstreetmap.org—was initially developed by Mapbox and DevelopmentSeed with funds from the Knight Foundation with the explicit purpose of improving core OSM infrastructure. Today, iD is a successful open-source software project, with more than 10,000 code commits on GitHub—whose core "maintainer" (that is, the lead developer) is a Mapbox employee. Other Mapbox-maintained tools include the OSM validation utility, OSMcha, and a number of OSM data-processing tools available for anyone working with OSM data. Bing is the primary imagery provider for OSM. Telenav maintains the website improveosm.org, which boasts the tagline: "Tools and Data from Telenav, built for the OpenStreetMap Community." Some of these data are pre-processed datasets of potentially missing features identified by machine learning on telemetry data. These are just a few examples of corporations participating in OSM in addition to their paid editing-teams. This is far from a comprehensive listing of which companies have contributed useful utilities to the project. Tracing the decade-long involvement of developers, their employers, and the variety of funding sources (corporate, donation, NGOs) is beyond the scope of this article, but it is safe to say that corporate involvement in OSM has shaped and maintained the project as it exists today.

Corporations also have access to rich geographic data from their customers and operations. For example, telemetry data (typically location data from mobile devices) can be used to identify missing roads, turn-restrictions, one-ways, and more. Mapbox compiles these data internally to assist their data-teams (mapbox.org/Telemetry). Amazon Logistics reports that they use their delivery driver's GPS traces in conjunction with driver feedback to help improve OSM [10]. Grab also states on their wiki page that their data-team process begins by downloading their internal GPS traces [69]. In terms of improving the OSM road network, there are few substitutes to such telemetry data. These datasets leverage a massive number of sensors, obtaining more ground-reference GPS traces than any number of individual OSM contributors could possibly acquire as hobbyists.

Some members of corporate editing teams are not new to OSM. At least 14 members of the corporate editing teams were actively editing before 2014 as individual contributors. Collectively, they represent 1% of all corporate editors, but their total edits to the map are equal to about 4% of all corporate edits since 2014, suggesting they are heavy corporate editors. In a 2017 OSM Foundation survey, 55% of respondents that were associated with an organization engaged in paid editing were with OSM for 3 years or more before joining said organization [65].

Corporations often sponsor and participate in OSM conferences. From our observations, their presentations are some of the best attended talks at State of the Map conferences, especially if it is the corporation's first talk to the OSM community. They can also be some of the most contentious, prompting aggressive remarks and questions from the community.

*3.2. Quantitative Evaluation Using Historical Quarterly-Snapshot OSM-QA-Tiles*

Next, we use historical quarterly-snapshot OSM-QA-Tiles to visualize and understand the global footprint of the 10 corporate editing teams and the features they are editing. Since mappers may have been active before being employed by a data team, knowing the date when a mapper becomes a corporate editor is an important detail. The resolution of this detail is limited to the responsiveness of the company itself to update their publicly-facing list of data-team employees. For example, if a mapper stops mapping as an employee in January, but the editor list is not updated on the wiki until February, there is no publicly observable method to know that any edits between these times were not corporate edits. At the moment, manually tracking and updating these lists for this type of research is possible, but as these teams continue to grow, this task will become too burdensome for individuals to do manually. Beyond measuring where corporate editors are active and what they are mapping, we show distinct temporal editing patterns that characterize these teams. Specifically, corporate editing teams appear to follow a Western five-day work week, where the activity of these teams is punctuated starkly with periods of little to no editing every five days by apparent weekends as days off. Later we discuss how corporate editing may be identified in the future expressly from these distinct temporal patterns.

### 3.2.1. Global Footprint

Figure 3 shows the global footprint of the 10 corporate editing teams. This map is produced by plotting the location of every edit by a member of a corporate team (denoted by color). High concentrations of edits appear to glow white. Together, the power of corporate editing is globally reaching. Mapbox and Apple have the largest footprints with edits on all six populated continents. Telenav and Amazon mostly edit in North America and parts of Europe whereas the Microsoft team is focused mostly on North America and Australia. Grab predictably focuses only on Southeast Asia. While Uber does have some edits all over the globe, they are primarily active in New Zealand. As mentioned earlier, Facebook's work is heavily focused in Thailand. Overall, Figure 3 shows that corporate editing is a global phenomenon with specific regions of more interest to some companies than others.

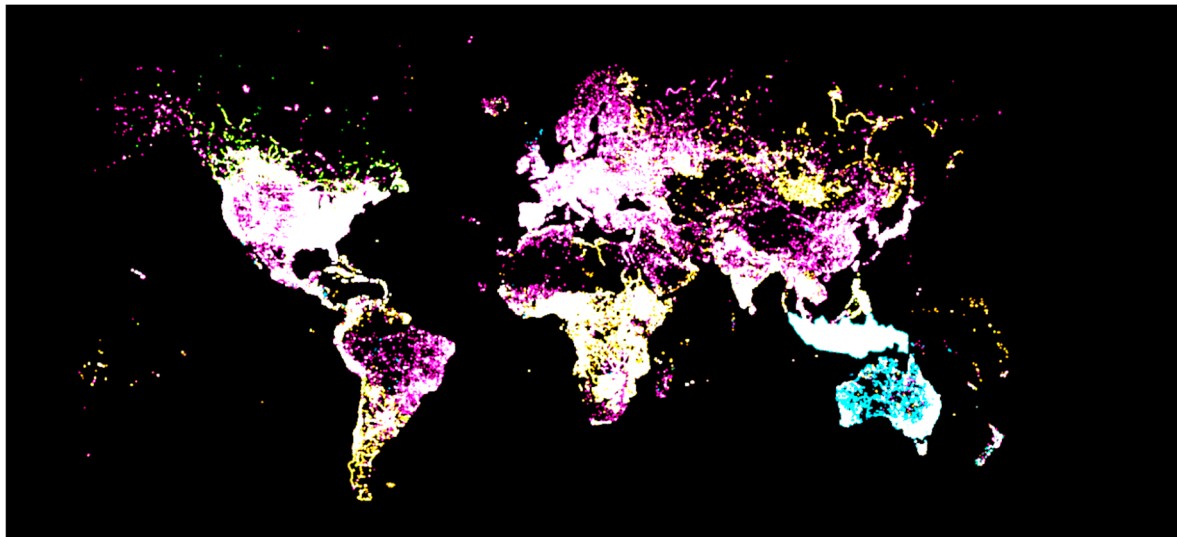

**Figure 3.** *Cont.*

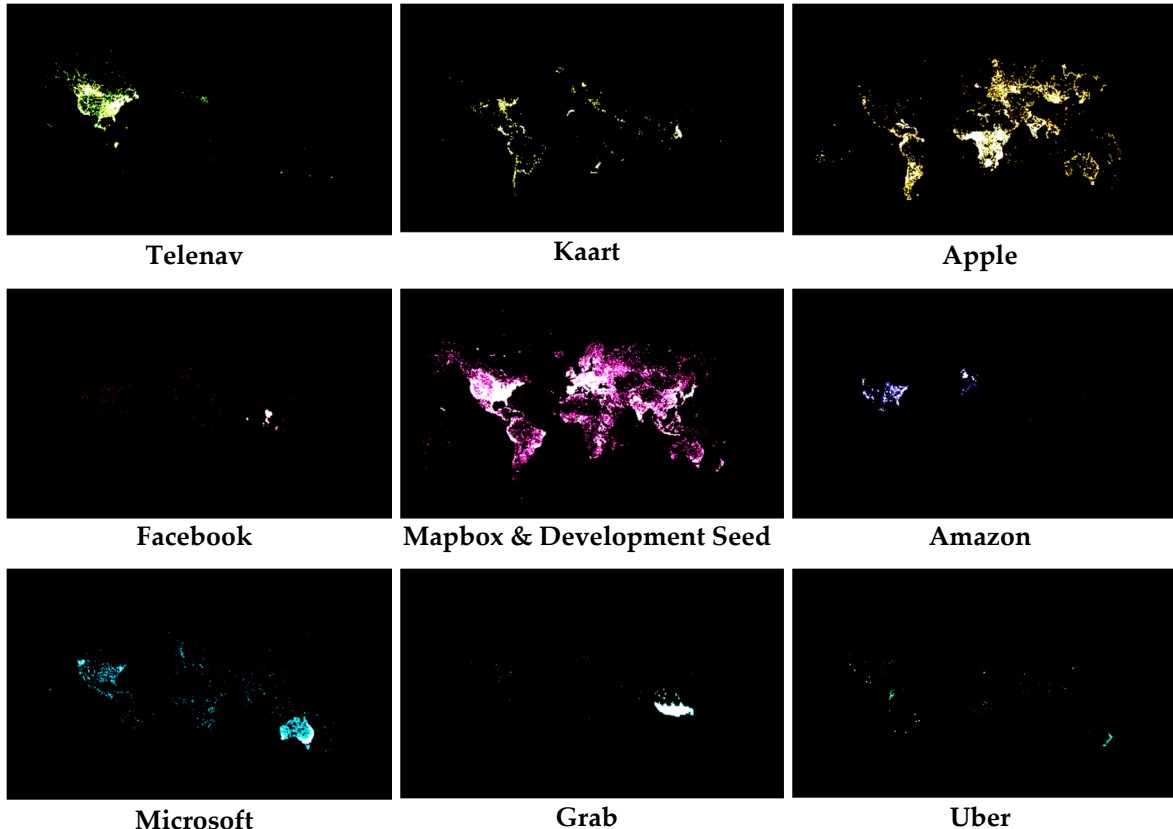

**Figure 3.** Where corporate editors are editing. The main map shows an aggregated view for all 10 companies. The sub figures show where each company is editing. In this map, we have combined the Mapbox and Development Seed teams because they merged in late 2017.

### 3.2.2. What Are Corporate Editors Mapping?

Table 2 highlights the increasing activity of corporate editors over the last 4 years. 2018 stands out as a remarkable year as it seems to indicate a change in collective focus towards editing road networks and building data. Figure 4 shows the relative quantity of edits to buildings, kilometers of road, points of interest, and amenities per team per year, compared to the total number of edits in the area. Thus, the radar charts highlight the main features of focus for the teams in the areas they are editing. For example, in 2018, Apple editors, on average, were responsible for nearly 80% of all the edits to existing roads and 70% of all the new roads created in the areas where they were active, defined by zoom level 12 map tiles (about 95 km$^2$ at the equator, the size of a small city). Generally, companies have a preference for a particular edit type. Telenav and Grab, which focus on navigation, are primarily editing roadways. In the case of both corporations, they are editing existing roads more often than they are creating them. Apple, Microsoft, and Facebook also have a massive imprint on the road networks in the areas their data teams are active. In 2017 and 2018, these teams were responsible for creating more than half of the new roads and editing more than half of the existing roads. Compared to all editors, Uber never dominates editing in regions where they have been active in the two years they have been involved in corporate editing. We see that in 2017, they were more focused on editing buildings, amenities, and points-of-interest; they did not focus on road editing until 2018. In recent years, Kaart continues to be responsible for over half of the total road edits in regions where they are active, but the percentage of buildings, amenities, and points-of-interest they are mapping has been decreasing, on average. Grab, which has only been active in the last year, has been predominantly mapping the roads in the areas in which they operate, making them responsible for nearly 75% of both new roads and edited roads in the map.

**Table 2.** Total number of features, kilometers of roads, number of buildings, amenities, and points of interest edited per year by all corporate editors. The increase in number of features edited since 2015 shows the overall rise in corporate editing.

| Year | Features | New KMs of Road | KM of Existing Roads | New Buildings | Edits to Existing Buildings | Amenities | POIs |
|------|----------|-----------------|----------------------|---------------|-----------------------------|-----------|------|
| 2015 | 1,703,107 | 96,604 | 660,591 | 321,535 | 47,730 | 13,892 | 40,096 |
| 2016 | 2,251,615 | 87,321 | 677,795 | 308,785 | 198,366 | 69,949 | 214,087 |
| 2017 | 3,121,727 | 179,256 | 591,627 | 632,859 | 305,665 | 58,616 | 178,887 |
| 2018 | 9,925,463 | 682,938 | 2,982,248 | 1,709,935 | 176,113 | 33,845 | 61,238 |

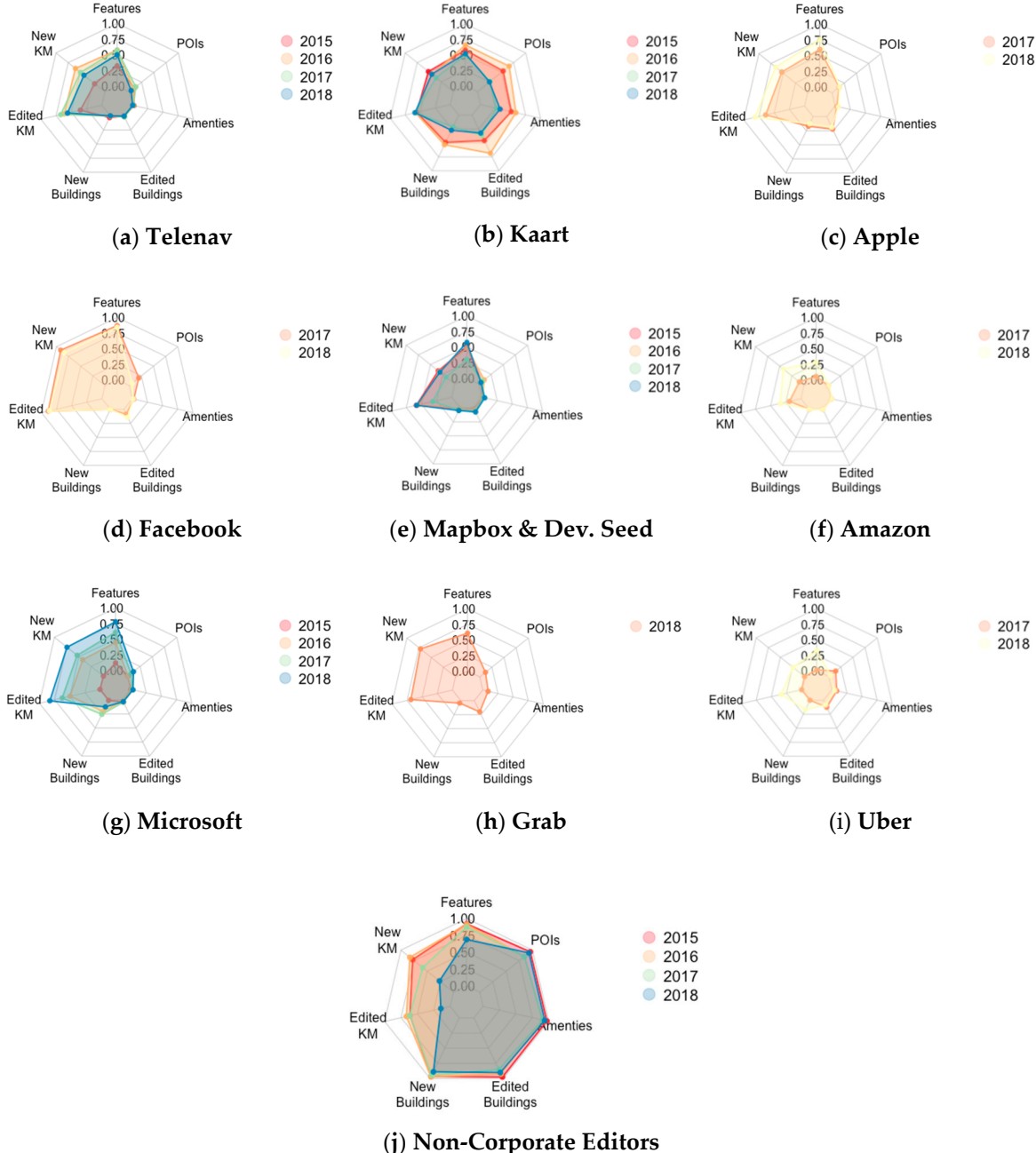

**Figure 4.** Each figure shows the types of edit these companies performing, relative to the total editing activity where they are active. These are annual averages over all of the zoom level 12 map tiles where a company is active. "Features" refers to editing any feature (all types of edits). The final figure (j) represents the activity of non-corporate editors in areas where (any) corporate-editors are active.

### 3.2.3. Characterizing Corporate Editing Patterns

Corporate editors also leave a distinct Monday through Friday time-signature in the database. In addition, Figure 5a shows the difference between corporate and non-corporate editors in terms of their lifespans of active editing. The solid line represents the number of editors starting from their first edit, while the dotted line represents the number of editors on the day for which they made their last edit. Depicted this way, the area between the two lines represents the size of the active community at any given time as people join and leave. This figure shows that even though there are less than 1000 corporate editors, the relative size of the active community is larger than the number of total mappers when OSM first began, and generally more stable in terms of ongoing contribution. The primary difference is that there are very few "one-time contributors" on corporate data teams; this one-time contribution behavior, which is more common in the general OSM community, drives the two lines closer together as the first edit and last edit of a mapper are on the same day. Instead, the slope of the dotted line is steep at the end of 2018, showing that many editors are still active right up to when we pulled the data.

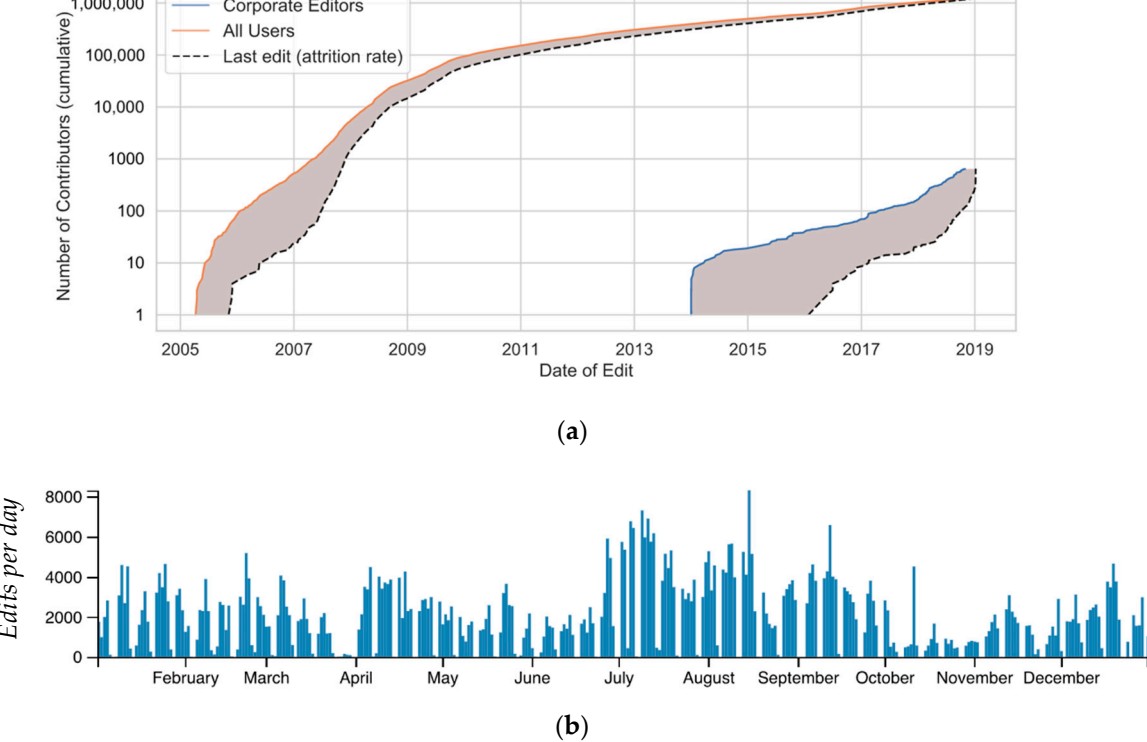

(**a**)

(**b**)

**Figure 5.** Characteristics of Corporate editors: (**a**) The rate of growth of all OSM editors compared to corporate editors. The solid lines represent number of contributors denoted by the day of their first edit. The dotted lines represent the number of users denoted by the day of their last edit. The shaded area between the solid lines and the dotted lines could be thought of as the relative size of the "active" community. These two lines converge at the end because those are the most recent edits in our data. The steep slope in the corporate-editors dotted line shows that these editors have been active recently (not one-time contributors); (**b**) Edits per day by the Facebook team in 2018. Consistent activity throughout the year showing 52 weeks of relatively consistent work five days of the week, with no editing on weekends. This pattern of consistent weekday editing is present across all of the data teams we have examined.

A signal of possible corporate editing is the apparent weekly pattern of editing activity expressed in Figure 5b. Each of the data-teams explored here maintain this same pattern: editing consistently during the week throughout the year with few-to-no edits on weekends. This also suggests that these

unique temporal signatures could be used to identify corporate editing activity by teams that have not yet disclosed a list of users or come forward publicly as using and contributing to OSM. This could be quite volatile if these editors are found to be in violation of the organized editing guidelines. Preliminary analysis identifies another 3000 active mappers that exhibit similar editing patterns in temporality and volume: many of whom are involved in import efforts and humanitarian mapping tasks. However, there is currently no evidence suggesting these are undisclosed corporate editors and, moreover, it is difficult to validate such accusations unless the mapper self-reports an employer in their user profile—as is common for currently known corporate editors.

## 4. Discussion

The growing phenomenon of corporate editing is the latest evolution of corporate involvement in OSM. Of specific interest is the massive growth in the number of corporate editors and the apparent investment that corporations are making in OSM. Though prolific, corporate editing varies in geographic reach, objects edited, and volume across corporations.

While Figure 3 may initially appear to present corporate editing as dominating the map, Table 2 and Figure 4 explores the impact of these edits. Though there is disproportionate impact across the globe, it appears that corporate editors have the largest impact on the road networks in areas where they are active (compared to buildings, amenities, points-of-interest). This is not a surprise given the value of a routable road network, but also not out of character for the evolution of the map without these editors: the map often evolves first from the road network [55]. This does raise further questions about longevity of corporate interest once the road networks are complete in these areas: will there be motivation for these corporations to map buildings or points-of-interest?

Figure 4j shows while there has been a consistent rise in corporate editing as a percentage of the total edits, non-corporate editors are still the dominant force who are responsible for nearly 70% of all features edited in 2018 (averaged globally in areas where corporate editors were active). Meanwhile, the percentage of the road network edited by non-corporate-editors is under 30% for these areas, on average. This means that corporate editing is having a significant impact in the regions where it is happening, but it is not currently dominating the *global* map. However, the motivations of corporate involvement and their long-term impact on the OSM data and community require further research.

In terms of motivations for mappers, Budhathoki and Haythornthwaite found that, among other factors, learning OSM to demonstrate proficiency to future employers was a potential financial benefit of contributing to OSM [7]. Published in 2013, this study predates the rise in corporate editing and found that while these financial benefits and career outcome were relatively low motivators for contributors, the notion of such financially motivated mapping was present. The increasing interest of various corporations (Table 1) and the growth of the number of corporate editors (Figure 5a) highlights the evolution of OSM and may change the motivations for contributors. With regards to OSM as a VGI platform, these corporate editors exacerbate the double-edged sword conundrum highlighted by Haklay and Seiber [70]. On one hand, compensation typically means some level of expert or professional involvement, indicating high quality data and validation. On the other, if contributions are paid for, the data can be seen as coerced, and perhaps even disqualify as VGI, taking away the benefits of crowd-wisdom and local knowledge for which VGI is recognized. While OSM contributions are still majorly volunteered (Figure 5a), the prolific activity of corporate editors pushes the threshold of OSM's status as a VGI project. Regardless of who contributes data, as long as the quantity and quality of the data improves, OSM will continue to be a valuable open data platform.

## 5. Conclusions and Future Research

In what we believe to be a first report of the phenomenon of corporate editors in OSM, we have highlighted corporate editors' place in the community and their visible footprint on the map. Our analysis addresses some of the current tension in the OSM community regarding this new phenomenon. The historical context and observational analysis also highlight the multi-faceted

involvement of the companies in OSM which go beyond just editing the map. Corporations appear to have their own map editing agendas that are probably aligned with corporate interests. We also note that other organized groups as well as individuals have been cited as having particular interests that drive their contributions to the map. The contributions are further shaped by the values embedded in the technology which drive who can participate and how [30,71]. Thus, the combined effort of all groups driven by their own set of values, motivations, and goals, mediated by the OSM platform produce what is perceived as the unified map of the world. With the ongoing growth and map spread of corporate editing in OSM, it is too early to draw conclusions about the lasting impact of this new iteration of corporate involvement through paid mappers. Instead, we raise questions for consideration about how OSM might evolve.

First, how does corporate editing activity affect the map data? We might wonder if we can separate ideologies of the sources of the data from its presence. One might argue that the uneven coverage of data in OSM and the large-scale edits that corporations are capable of making can close this gap in the database. Additionally, the data added by corporate editors will probably be of good quality because these editors are trained, have economic incentive and managerial oversight, and because editing in these areas will bring attention to the map via a variety of edit-monitoring services. Data are more likely to improve in areas where some data already exists [72,73]. Thus, these activities, especially in developing nations, may be looked on as *map seeding* which prompts growth of the OSM community and *densification* of the data. In developed nations, the editing activities are probably going towards progressive data updates and quality improvements, and thus are more in line with *map gardening* [74]. However, another important argument comes from the point of view of bias toward self-serving interests: are corporations introducing geographic bias into the map? As the map continues to be filled in, will corporate interests have too much voice in what and where gets mapped?

Second, how does corporate editing affect local communities? Historically, the attitude towards large corporations have been contentious with avid mappers being more congenial [7]. One concern is that corporate editing is squeezing out existing "local" mappers. The organized editing guidelines advocate strongly for working with local communities to avoid this. While the data shows that corporate editing is certainly prolific and is found to be the largest editing force in many places, it is unclear what the relationship is or may become with local mapping communities. Empirically, we observed through the wiki and Github repositories that the reported corporations are currently cooperating with the organized editing guidelines and reaching out to local mapping communities. The community has also arranged itself in such a way as to monitor if corporations overstep in ways that the community can currently foresee, but as the OSM landscape evolves, additional mechanisms might need to be put in place.

Third, are corporations acting reciprocally with the OSM community, and offering as much or more as they are getting from their OSM involvement? Some corporations have access to large, rich datasets (telemetry) that no one else has, which could in turn improve the map if shared—but how much do corporations share? It says something about the value of geospatial data when we observe that achieving a more complete map is driving corporations to collaborate. Furthermore, global-scale validation and monitoring is difficult for individuals because of the sheer volume of edits. We know from conference presentations and the production of tools that corporate editors actively monitor map changes for vandalism and accuracy at scales beyond the abilities of individual contributors.

Fourth, what is the best way for the community to monitor and support corporate editing, assuming that it does have collective value? Mechanisms put in place such as the organized editing guidelines are primarily based on self-reporting, which is what assessment is then based. However, as the number of corporate entities continues to grow, maintaining lists of usernames and corresponding edits could become onerous. Further mechanisms may be needed so that the community can hold corporate editors accountable, ensuring that (1) their community engagement is proportional to their impact and subsequent benefits from the data, and (2) that their impact is constructive, and in keeping with shared goals of the OSM community.

Ultimately, consequences that stem from the publicized activity of corporations' data production might have yet different effects on market and corporate behavior. Having quantified and contextualized the current footprint and involvement of corporate editing, our hope is that new research about OSM can arise as this vibrant community continues to evolve.

**Author Contributions:** Conceptualization, Jennings Anderson, Dipto Sarkar, and Leysia Palen; Methodology, Jennings Anderson and Dipto Sarkar; Software, Jennings Anderson; Validation, Jennings Anderson and Dipto Sarkar; Formal Analysis, Jennings Anderson and Dipto Sarkar; Data Curation, Jennings Anderson and Dipto Sarkar; Writing-Original Draft Preparation, Dipto Sarkar and Jennings Anderson; Writing-Review & Editing, Leysia Palen, Jennings Anderson, and Dipto Sarkar; Visualization, Jennings Anderson and Dipto Sarkar.

**Funding:** This work is made possible by the U.S. National Science Foundation, Grant IIS-1524806.

**Acknowledgments:** The authors thank colleagues both at the University of Colorado Boulder and in the OSM community for their valuable feedback and deeper understanding of the long, nuanced history of corporate involvement in OSM. Additionally, we thank the various custodians of data-team user lists that made conducting this research possible and their willingness to help gather and curate a master list of corporate editors. Additional thanks to colleague Kenneth M. Anderson and the U.S. National Science Foundation for funding support.

**Conflicts of Interest:** Anderson was a research fellow with Mapbox in 2016 and 2017 but is not currently employed by the company.

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
