# Peer review of "Corporate Editors in the Evolving Landscape of OpenStreetMap"

_ijgi, doi:10.3390/ijgi8050232_

Round 1

Reviewer 1 Report

Corporate Editors in the Evolving Landscape of OpenStreetMap

The paper addresses a very timely and pressing issue for the OSM community. There is plenty of speculation as to the corporate influence in map editing, yet little actual quantitative research on this area. Indeed, the paper presented may be one of the first, outside of conference presentations. The importance of publishing this work is high.

Abstract

Contains all required components - clearly gives a brief of the research conducted and main findings.

Keywords - I am not sure that neogeography has much resonance these days. VGI would be more appropriate, or perhaps crowdsourcing. On its own 'global' doesn't mean anything as a keyword. 'Open Data' may be useful here too.

Introduction

First sentence is a bit awkward. How about 'OpenStreetMap was founded in 2004 with the express goals of....'. Just thinking about how tailor the paper to a broader audience. Might want to mention Steve Coast, Ordnance Survey, and ref his book on OSM, just to tie this part up.

I believe that the characterization of OSM as a 'communitiy of communities' is spot on. Is this a direct quote from another source, or an original term that you've put together? If it is borrowed, you may want to reference that source, even if it is a general description used by the community itself. This is just a minor point.

Page 2 line 51. Can you quantify the major growth? Like in terms of contributors? Or I suppose this is followed up later.

Page 2 line 56. What is Mapbox? I personally know what Mapbox is, but the average person reading this may not.

1.1 OSM contributors

Can you better characterize the organizational structure of OSM? The role of the OSMF and how HOT is not actually part of OSMF. Again, a minor point, but I think you want to make sure that you are crystal clear about these things for the sake of the OSM community.

Your discussion of the number of map contributors is great, same with the reproduction of offline inequalities. I like Figure 1 and (a) (b), however you present these before dicussing  how you obtained/extracted the data....I know you get to it later, but the order needs to be re-jigged. This isn't literature review if you are presenting new data.

1.2 Landmark corporate and government contributions to OSM.

This is very well done and gives excellent context to the level and degree of corporate and gov involvement in OSM.

Materials and Methods

Some typographic and grammar issues on Page 5 lines 187-189.

Interesting data gathering approach using quarterly snapshots. Obviously you are not proposing that this is the complete set of data created or edited by corporate editors (since it would be impossible to truly discern those individuals anyways, other than through public and voluntary disclosure by those companies). I do think there is space here for a diagram showing this data gathering method (the OSM-QA tile part). Just to clarify the process a bit. Regardless, this is a solid methodology and I am supportive.

Table 1 is fantastic. This is a great resource to have in one place (though as you note, it is only a snapshot.)

Results

3.1 Observational analysis of corporate involvement

3.1.1 Tracing corporate interest through media

These companies that you discuss (geofabrik, graphhopper) aren't the same as in table 1. I think this part may be getting a bit off topic, since they are not the corporations doing large scale editing (or are they? We don't know, I guess). I would be tempted to delete page 7 lines 222-229 and just focus on the corporate entities identified in Table 1.

Page 8 lines 272-280. Very interesting that you mention the very fresh issues of GlobalLogic. I wonder if this is too recent an issue to discuss in the manner that you are? It seems a bit too speculative for the purposes of this paper. It also risks shifting the focus on this paper away from the good work that you have actually done with simply quantifying the contributions from corporate actors - I don't think you need to speculate on motivations or the outsourcing supply chain at this point.

3.1.2 Contributions to the larger ecosystem and community participation

Page 8 line 311-312. Some speculation of how/why people got hired at specific companies. I am not sure that this is a critical part of your paper. You make a good case in this section for how corporations support OSM in other ways that edits. You want to make sure that this important work isn't perceived as being speculative. This qualitative section has a tendency to wander that way vs. The quantitative section.

3.2 Quantitative evaluation

Global footprint of the 10 corporate editing teams. I think this is the real strength of the paper and it answers the research questions posed. The challenge of discerning between corporate and personal edits is a tricky one, as you note, but I do not think this invalidates the work at all, but is rather a caveat. The second paragraph (page 9, lines 330-333) is a fragment, I believe you should either expand this into a full paragraph or merge with the previous quantitative introduction component.

3.2.1 Global Footprint

Figure 3. I do like the aggregate map, however this is difficult to parse, and also begs the question if this overall footprint is meaningfully different than the non-corporate footprint? Looking at the footprint broken down by corporation is much more useful. The description of the geographic areas for each company is fine, but I find it somewhat conversational.

3.2.1 What are corporate editors mapping?

Page 10 line 363 "Kaart has been less dominatly...". Very awkward, please re-word this sentence.

You might want to define what zoom level 12 covers (in general - just to give the reader an idea of what they would expect to see at this level).

Figure 4 is fantastic. I really got quite a bit of information out of this, though it is difficult to compare given the varying levels of total edits done by each company. What is clear is that most of the activity is creation of new features and editing of roadways. Other than Kaart, I was surprised at the low levels of POI editing, since I assumed that this richer attribute data would be more in line with corporate interests.

3.2.3 Characterizing Corporate Editing Patterns

The information shown in Figure 5a is really interesting, but I think it needs to be explained in a more direct manner. I see the gap between the total number of contributors and their last edit, but the joining of the tail at the most recent time step needs to be better explained to the audience. Also, it may be beneficial to shade or otherwise highlight the area in between the two lines?

Discussion

First paragraph is a fragment, incorporate this into the second paragraph.

Pge 12 line 413 "corporations appear to have their own map editing agendas that are probably alighened with corporate interests"...for example. I have no doubt that this is true, but you should highlight what has led you to this conclusion a bit more forcefully. Also, the subsequent point that no one is likely editing OSM without their own particular agenda (either geographical, topical, etc.) is made. This is perhaps a good note to being this section on - admiting that all who edit the map or are involved with OSM are enacting their own personal agenda (there is some literature from Haklay that could be fed into support this - more citation in this section would be helpful in general).

Page 12, line 421-422. How OSM might continue to function as the definitive instance of an open geospatial data base. Careful here, first, you should probably step back and consider exactly what you've defined as being 'open' in this context. Because corporations contribute greatly to OSS development (like Linux, for example) does it make the software less open or otherwise usable? I guess I'm just confused by this statement, since it seems to imply that corporate editing would possibly make OSM less open...but then you haven't really defined what 'open' means in this instance, or what the OSM interpretation of 'open' would be. So I'll pose a question - isn't the goal of OSM to have more data, regardless of who contributes it? I also don't think you need to answer this in your paper, you just need to play the middle of the road and point out that there are edits coming from different communities, and your research is the first to really start to try and break out those communities along some aggregate lines, based on the user information volunteered and patterns you present. This actually leads into a general comment that I have been thinking about how to make for a while - have you considered the impact of this paper on the OSM community itself? Like are hobby mappers going to raise pitchforks and burn corporations in effigy as a result of this? Maybe, but I think it is very important to consider how you frame some of the statements in this discussion and even throughout the paper. There are a few times that you frame it as a study on the increase or influence of corporations within OSM. Obviously you are free to do so, but I would suggest framing this around defining communities of interest within OSM, and the most evident one is corporations. This is exploratory work and you don't want to push the discussion points past what you've got data to support. So while I personally agree with your discussion points about map seeding, but the hypothetical questions you pose about corporate interests and voice about what gets mapped and where starts to read a bit too much like the OSMF-talk mailing list. This is also true in the second point about how corporate editing affects local communities. This is super controversial, as you probably know, and you don't really provide any data that discusses this impact (you didn't study the effects on local communities - this would be a great follow up study however). This shouldn't be in the discussion (where you build your findings on top of previous research), but rather as a point for future research in the conclusions - so how does this corporate mapping that you've identified impact local communities? Maybe it is welcomed in some places, maybe it is totally unnoticed in others, etc. There is a lot of nuance here that is needed to break out the broad brush you are using in this section.  Each of these discussion points (1-4) would be best as future-facing questions in the conclusion - they are statements that beg future research rather than showing your results in context.

I think that this discussion, given the academic publication venue, would be much stronger if pulled back to contributions made by your research that build on the general area of OSM studies or even open software literature. Keep the speculative parts for a future research section in the conclusion.

Conclusions.

I think an easy fix here is to just copy/paste the discussion into the conclusions and then write a discussion that contextualizes your research findings v.v. Previous research on OSM communities.

Overall comments.

This is a strong paper on a very important topic. Overall, I am very supportive of this paper and think it has potential to make a large impact in the area of OSM studies and VGI. I do think that the discussion section is too speculative and needs to go into the conclusion as a type of next steps blueprint. The questions you ask there are not answered with any data, so they don't belong in the discussion. Use the discussion to present the meaning and significance of your work in the broader context of VGI/crowdsourcing and open databases. How open is open if corporations are contributing? Is OSM volunteered if this % of contributions are paid for? There is some good literature out there for you to contribute to, and many previous OSM studies to connect with (Monica Stephens, Nama Budhathoki, Muki Haklay's work - you already cite these, but you should tie in with their work in the discussion).

Author Response

The paper addresses a very timely and pressing issue for the OSM community. There is plenty of speculation as to the corporate influence in map editing, yet little actual quantitative research on this area. Indeed, the paper presented may be one of the first, outside of conference presentations. The importance of publishing this work is high.

Thank you for your review, we agree that this is timely, important research to be conducting in the OSM and VGI space. We know of one talk given at the 2019 FOSSGIS conference in Dresden, Germany that touches on the subject, but there is no other written, published work to date.

Keywords - I am not sure that neogeography has much resonance these days. VGI would be more appropriate, or perhaps crowdsourcing. On its own 'global' doesn't mean anything as a keyword. 'Open Data' may be useful here too.

These are valuable suggestions about the keywords which we agree with and have changed appropriately.

First sentence is a bit awkward. How about 'OpenStreetMap was founded in 2004 with the express goals of....'. Just thinking about how tailor the paper to a broader audience. Might want to mention Steve Coast, Ordnance Survey, and ref his book on OSM, just to tie this part up.

We have rephrased the beginning to reference the Ordnance Survey and cite his book.

I believe that the characterization of OSM as a 'community of communities' is spot on. Is this a direct quote from another source, or an original term that you've put together? If it is borrowed, you may want to reference that source, even if it is a general description used by the community itself. This is just a minor point.

This phrase is not yet commonly used, I first heard it from Professor Patricia Solis during a talk about the YouthMappers project at the AAG Annual Meeting. We cite this talk at the end of the sentence when it is first introduced.

Page 2 line 51. Can you quantify the major growth? Like in terms of contributors? Or I suppose this is followed up later.

The growth is quantified first in Table 1: Those numbers represent edits since 2014. These teams did not exist before that. We have also added a new Table (Table 2) that shows annual counts for all corporate edits, broken down by edit type.

Page 2 line 56. What is Mapbox? I personally know what Mapbox is, but the average person reading this may not.

We were consciously avoiding giving preferential treatment to Mapbox over other companies and defining/defending them in the context of OSM. If someone is unsure about Mapbox, we trust they can look this company up themselves.

Can you better characterize the organizational structure of OSM? The role of the OSMF and how HOT is not actually part of OSMF. Again, a minor point, but I think you want to make sure that you are crystal clear about these things for the sake of the OSM community.

A minor point, but it’s importance is certainly understood and noted, we have added a few sentences to better describe this.

Your discussion of the number of map contributors is great, same with the reproduction of offline inequalities. I like Figure 1 and (a) (b), however you present these before discussing  how you obtained/extracted the data....I know you get to it later, but the order needs to be re-jigged. This isn't literature review if you are presenting new data.

These figures were actually calculated from an instance of the OSM changeset database, not the OSM-QA-Tiles (our methodology for the rest of the paper). We have added a description of this to the captions. Yes, these numbers are ours (as in not published elsewhere, calculated by us), but they certainly are not novel. As these numbers are ever changing, we think it is worthwhile to present the latest, most accurate numbers here.

Some typographic and grammar issues on Page 5 lines 187-189.

Thank you for identifying these, we have fixed them.

Interesting data gathering approach using quarterly snapshots. Obviously you are not proposing that this is the complete set of data created or edited by corporate editors (since it would be impossible to truly discern those individuals anyways, other than through public and voluntary disclosure by those companies). I do think there is space here for a diagram showing this data gathering method (the OSM-QA tile part). Just to clarify the process a bit. Regardless, this is a solid methodology and I am supportive.

We are glad to have your support with this methodology. Unraveling the history of OSM through the data is a convoluted and difficult process with many pitfalls. These difficulties are exacerbated when doing a global analysis. Developing these methodologies has been years in the making. There is a footnote in this section that links to a technical description of how the historical quarter OSM-QA-Tiles are made. We have also added a sentence and reference to another paper that more deeply explains this analysis workflow.

These companies that you discuss (geofabrik, graphhopper) aren't the same as in table 1. I think this part may be getting a bit off topic, since they are not the corporations doing large scale editing (or are they? We don't know, I guess). I would be tempted to delete page 7 lines 222-229 and just focus on the corporate entities identified in Table 1.

We agree, it is simplest to just delete these lines and focus on the known companies. Furthermore, another reviewer was confused by this metric of engagement, so we are choosing to only focus on what we know in this section: these 10 companies.

Page 8 lines 272-280. Very interesting that you mention the very fresh issues of GlobalLogic. I wonder if this is too recent an issue to discuss in the manner that you are? It seems a bit too speculative for the purposes of this paper. It also risks shifting the focus on this paper away from the good work that you have actually done with simply quantifying the contributions from corporate actors - I don't think you need to speculate on motivations or the outsourcing supply chain at this point.

We agree these are speculative in nature, but our purpose here is simply to comment on the presence of such speculation to highlight the skepticism of corporate involvement. Another reviewer found this commentary to be helpful background.

Page 8 line 311-312. Some speculation of how/why people got hired at specific companies. I am not sure that this is a critical part of your paper. You make a good case in this section for how corporations support OSM in other ways that edits. You want to make sure that this important work isn't perceived as being speculative. This qualitative section has a tendency to wander that way vs. The quantitative section.

We have deleted this line because it was purely speculative as framed. While there is truth to it and examples can be found in Steve Coast’s book, you are right, this distracts from the main points of the paper.

Global footprint of the 10 corporate editing teams. I think this is the real strength of the paper and it answers the research questions posed. The challenge of discerning between corporate and personal edits is a tricky one, as you note, but I do not think this invalidates the work at all, but is rather a caveat. The second paragraph (page 9, lines 330-333) is a fragment, I believe you should either expand this into a full paragraph or merge with the previous quantitative introduction component.

We expand on this later in Section 3.2.3, so we merged it with the previous paragraph.

Figure 3. I do like the aggregate map, however this is difficult to parse, and also begs the question if this overall footprint is meaningfully different than the non-corporate footprint? Looking at the footprint broken down by corporation is much more useful. The description of the geographic areas for each company is fine, but I find it somewhat conversational.

The global map makes it clear how pervasive corporate activity is as a whole (and yes, it looks somewhat similar at this resolution to the non-corporate editing map). This is important because we are positioning corporate entities as one of the communities in the ‘community of communities’ description. Once the larger picture is presented, it can be further broken down to the individual companies.

Page 10 line 363 "Kaart has been less dominately...". Very awkward, please re-word this sentence.

Done.

You might want to define what zoom level 12 covers (in general - just to give the reader an idea of what they would expect to see at this level).

Done. Zoom 12 tiles are about 95 sq. km. at the equator, about the area of a small city, w have added this as a footnote.

Figure 4 is fantastic. I really got quite a bit of information out of this, though it is difficult to compare given the varying levels of total edits done by each company. What is clear is that most of the activity is creation of new features and editing of roadways. Other than Kaart, I was surprised at the low levels of POI editing, since I assumed that this richer attribute data would be more in line with corporate interests.

Thank you, and we agree: these patterns were interesting to see. One caveat of these radar charts is that they represent percentage of total activity, not raw values. For this reason the edit  data counts in Table 1 and Table 2 show the volume of edits while these radar charts show their relative impact in the regions they are active. We were also a bit surprised to see that across the board roads and buildings are more important to these corporations. Perhaps these corporations have other data sources for POIs/Amenities?

The information shown in Figure 5a is really interesting, but I think it needs to be explained in a more direct manner. I see the gap between the total number of contributors and their last edit, but the joining of the tail at the most recent time step needs to be better explained to the audience. Also, it may be beneficial to shade or otherwise highlight the area in between the two lines?

We have shaded the space between these lines and further described the nuances of these figures both in the caption and in the text.

Discussion

First paragraph is a fragment, incorporate this into the second paragraph.

Done (and changed in accordance with next suggestions).

Page 12 line 413 "corporations appear to have their own map editing agendas that are probably aligned with corporate interests"...for example. I have no doubt that this is true, but you should highlight what has led you to this conclusion a bit more forcefully. Also, the subsequent point that no one is likely editing OSM without their own particular agenda (either geographical, topical, etc.) is made. This is perhaps a good note to being this section on - admitting that all who edit the map or are involved with OSM are enacting their own personal agenda (there is some literature from Haklay that could be fed into support this - more citation in this section would be helpful in general).

Our new discussion incorporates more of this work.

Page 12, line 421-422. How OSM might continue to function as the definitive instance of an open geospatial database. Careful here, first, you should probably step back and consider exactly what you've defined as being 'open' in this context. Because corporations contribute greatly to OSS development (like Linux, for example) does it make the software less open or otherwise usable? I guess I'm just confused by this statement, since it seems to imply that corporate editing would possibly make OSM less open...but then you haven't really defined what 'open' means in this instance, or what the OSM interpretation of 'open' would be. So I'll pose a question - isn't the goal of OSM to have more data, regardless of who contributes it? I also don't think you need to answer this in your paper, you just need to play the middle of the road and point out that there are edits coming from different communities, and your research is the first to really start to try and break out those communities along some aggregate lines, based on the user information volunteered and patterns you present.

This is certainly an important question that should be investigated in more detail, but perhaps not here. We have re-worded this sentence to have side-step this implication and simplify this segue into the questions for future research (the main takeaways). Where possible, we have tried to follow this advice of ‘playing the middle of the road.’

Furthermore, Is the goal of OSM to have more data, regardless of who contributes it? This is a powerful question that requires backing up: Does OSM have a single goal regarding the map data? I can think of many people who would join one side or the other on this issue.

This actually leads into a general comment that I have been thinking about how to make for a while - have you considered the impact of this paper on the OSM community itself? Like are hobby mappers going to raise pitchforks and burn corporations in effigy as a result of this? Maybe, but I think it is very important to consider how you frame some of the statements in this discussion and even throughout the paper. There are a few times that you frame it as a study on the increase or influence of corporations within OSM. Obviously you are free to do so, but I would suggest framing this around defining communities of interest within OSM, and the most evident one is corporations. This is exploratory work and you don't want to push the discussion points past what you've got data to support. So while I personally agree with your discussion points about map seeding, but the hypothetical questions you pose about corporate interests and voice about what gets mapped and where starts to read a bit too much like the OSMF-talk mailing list. This is also true in the second point about how corporate editing affects local communities. This is super controversial, as you probably know, and you don't really provide any data that discusses this impact (you didn't study the effects on local communities - this would be a great follow up study however). This shouldn't be in the discussion (where you build your findings on top of previous research), but rather as a point for future research in the conclusions - so how does this corporate mapping that you've identified impact local communities? Maybe it is welcomed in some places, maybe it is totally unnoticed in others, etc. There is a lot of nuance here that is needed to break out the broad brush you are using in this section. Each of these discussion points (1-4) would be best as future-facing questions in the conclusion - they are statements that beg future research rather than showing your results in context.

First, yes, we have certainly thought about how this paper will be received in the OSM community. We intentionally choose to not be cynical of corporate-editing because, as we declare in the paper, there is not enough evidence to suggest that OSM is being over-run. Furthermore, the long-time history of corporate involvement in OSM, which we have highlighted in this paper, makes these a complex issue (the equal-and-opposite cynical approach would say that OSM has always been corporate-vested). To this end, we have tried to be the middle-of-the-road, objective, first official reporting of the growing number of corporate-editors (a new form of corp. involvement).

Yes, the discussion about corporate editing impact on local communities is very controversial and is the logical next extension of this line of research. We chose not to dive into this issue here because it will be an entirely different line of questions which should be supported with a different methodology: Ideally involving a close analysis of the correspondences between corporations and local communities. At this point in the paper, it feels out of scope, so we instead propose it as future research for the community.

I think that this discussion, given the academic publication venue, would be much stronger if pulled back to contributions made by your research that build on the general area of OSM studies or even open software literature. Keep the speculative parts for a future research section in the conclusion.

This suggestion makes a lot of sense and we have moved our speculative future research questions into the conclusion.

Conclusions.

I think an easy fix here is to just copy/paste the discussion into the conclusions and then write a discussion that contextualizes your research findings v.v. Previous research on OSM communities.

Thank you, this is a great idea, we have done this and renamed it “Conclusion and Future Research”

Overall comments.

This is a strong paper on a very important topic. Overall, I am very supportive of this paper and think it has potential to make a large impact in the area of OSM studies and VGI. I do think that the discussion section is too speculative and needs to go into the conclusion as a type of next steps blueprint. The questions you ask there are not answered with any data, so they don't belong in the discussion. Use the discussion to present the meaning and significance of your work in the broader context of VGI/crowdsourcing and open databases. How open is open if corporations are contributing? Is OSM volunteered if this % of contributions are paid for? There is some good literature out there for you to contribute to, and many previous OSM studies to connect with (Monica Stephens, Nama Budhathoki, Muki Haklay's work - you already cite these, but you should tie in with their work in the discussion).

We have written a new discussion to address these points.

Thank you for your detailed and constructive review. We agree with all of your suggestions and have reworked the paper to incorporate them as best we could. One angle we tried to carefully attend to with this paper was to not be immediately cynical of corporate-editing and instead tie it into the larger history of OSM. We will not be surprised to see more cynical takes on paid editing begin to appear, but these feel base-less and agenda-pushing given the long history of corporate involvement in OSM. There is so much future work to be done to dig deeper into this new phase of OSM without grabbing the pitchforks: time will tell.

Reviewer 2 Report

Well written article.

One detail in line 246 ff: In fact it is just the other way round. It was the community who has ensured that the road network is navigable. Pls. recheck and rewrite that claim.

The only aspect to be enhanced IMHO content wise, is the first part of the summary in chapter "4. Discussion":

It's a good idea to "raise questions about how OSM might continue to function as the definitive instance of an open geospatial data base." (line 421).

But before that, the "Corporate editing Patterns" should be discussed much more explicitly than now e.g. by enumerating the companies and commenting their patterns.

Author Response

Thank you for your review. We have addressed your two comments below and the new version of the document includes the mentioned changes with “track-changes” on so you can see the difference.

One detail in line 246 ff: In fact it is just the other way round. It was the community who has ensured that the road network is navigable. Pls. recheck and rewrite that claim.

Thank you for catching this. We intended to say that a lot of road data came from government and corporate sources initially. We have rephrased it to more accurately reflect the initial import followed by the meticulous work of the community to clean it up and make it navigable.

The only aspect to be enhanced IMHO content wise, is the first part of the summary in chapter "4. Discussion": It's a good idea to "raise questions about how OSM might continue to function as the definitive instance of an open geospatial database." (line 421). But before that, the "Corporate editing Patterns" should be discussed much more explicitly than now e.g. by enumerating the companies and commenting their patterns.

We have re-written the discussion to include more of this discussion as supported by our data. This includes enumerating these companies and commenting on their patterns. We moved the speculative, future-looking questions from the discussion to the conclusion, per suggestions from Reviewer 1.

Reviewer 3 Report

The paper is very well written using excellent language. It is easy to understand it even for a reader that is not an expert in the field. The theme is somewhat interesting given that this angle hasn't been researched yet in the OSM community. The presented results of the research are clear and informative.

But what would be really interesting to see are examples and comparisons. Although sections 3.1.1., 3.1.2. 3.2.1.and 3.2.3 somewhat cover the following suggestions, I could account at least couple of items that would significantly improve the paper:

 Are there observable patterns for corporate editing except locations where they occur? For example, if Apple's contributions are observed, are all mapped roads done in the same fashion, using high or low estimated accuracy (many nodes or fewer nodes), always the same or different thematic attributes. I would expect that such corporate editing has some rules in order to make data more uniform so that more advanced algorithms can be used over mapped data.

For example, when roads, mapped by Apple and Amazon are compared, who does more detailed job, even if the conclusion would come from couple of examples?

Majority of the corporate OSM mappers have had significant experience with OSM before they became members of such teams. Which company tries to hire the most of experienced editors. It would be an important information, are those the big companies like Amazon and Apple which are not primary OSM users or those are companies that base their business on OSM data like Mapbox and GraphHopper?

Even though locations of corporate edits are important part of the research, sections 3.2.1. (the second one, it should be 3.2.2.) and 3.2.3., and Figure 4 appear to be the core of the research. What is missing is the nature of the edits. Do they add tags and leave out geometry as is or only fix geometries? It doesn't have to be quantified data, giving the estimation of such edits would be very important. Also, are there examples where corporate edits are fixed by regular volunteers (for example those that were never part of the corporate teams). I can see that authors have made a small disclaimer that it is hard to track changes in the corporate editing teams but I am sure that some illustrative examples could be found.

Table 1 gives comparison of the sizes of teams and the number of their contributions. It does not say the years of active contributions and it does not compare the numbers with number of total OSM edits. The Figure 4 does that, but for the areas where companies are mapping. Given that almost the whole world is is covered by the edits it seams that corporate edits are overtaking the OSM. How does that compare to global number of edits/new objects/existing objects? That would be very informative, especially for a naive reader. Maybe even giving some trends to see if corporate editing is literally overtaking traditional voluntarism. That would also give some insights into voluntary contributions motivations eg. "Do mappers try to build up mapping experience so they can be employed like professional OSM mappers?".

Even though very well written, without covering at least some of the previously mentioned topics and diving deeper into the data, the paper itself might be more appropriate for a conference than for a peer reviewed journal. Some of the new text could be used to replace the somewhat lengthy four pages introduction section. With some of the suggested improvements I would be happy to move this paper forward.

I was able to spot only one tiny error on line 422 "...OSM might continue to function as the definitive instance of an open geospatial data base." Writing is excellent.

Author Response

Thank you for your review. We have addressed your comments point-by-point below and the new version of the document includes the mentioned changes with “track-changes” on so you can see the difference.

Are there observable patterns for corporate editing except locations where they occur? For example, if Apple's contributions are observed, are all mapped roads done in the same fashion, using high or low estimated accuracy (many nodes or fewer nodes), always the same or different thematic attributes. I would expect that such corporate editing has some rules in order to make data more uniform so that more advanced algorithms can be used over mapped data. For example, when roads, mapped by Apple and Amazon are compared, who does more detailed job, even if the conclusion would come from couple of examples?

This is an interesting question with its own potential research agenda: company v. company. Particularly, could company-specific editing activities be meaningfully extracted, if so, could they be used to train ML models? Though it seems too speculative without insider-corporate knowledge whether or not these editors are mapping in a specific way to train algorithms down the road.

At this point, we do not think that there is enough known information about corporate-editing to start comparing the quality of one company’s work to another without more context. Since each company is focusing on different projects in different regions, this comparison does not seem rigorous. Though there are certainly great questions about impact and quality to be answered in this line of investigation, especially if we can find a specific job or task where corporations are naturally competing in OSM (and what would that mean?).

We have, however, added Table 2 that gives a better breakdown of the editing totals and growth per year by corporate editors.

Majority of the corporate OSM mappers have had significant experience with OSM before they became members of such teams. Which company tries to hire the most of experienced editors. It would be an important information, are those the big companies like Amazon and Apple which are not primary OSM users or those are companies that base their business on OSM data like Mapbox and GraphHopper?

We agree, this is a very interesting question, but our methodology is not set up to answer this question (well).  While we can identify a number of mappers that were active in the community before through their username, this is only possible if they use the same username once they get hired. Companies like Facebook have specific usernames, such as VL0001 and VL0002 that employees use when editing. We cannot track if the individual behind this username was an active mapper before joining from the data alone.  Furthermore there are many other ways to be familiar with OSM data other than map editing that could make someone valuable to a company data-team. This question would best be answered with surveys and interviews with actual data-teams than speculated from the data. Per another reviewer’s suggestion, we have removed the speculation in this sentence in the paper and instead only report the known data. We hope future research will do these surveys.

Even though locations of corporate edits are important part of the research, sections 3.2.1. (the second one, it should be 3.2.2.) and 3.2.3., and Figure 4 appear to be the core of the research. What is missing is the nature of the edits. Do they add tags and leave out geometry as is or only fix geometries? It doesn't have to be quantified data, giving the estimation of such edits would be very important. Also, are there examples where corporate edits are fixed by regular volunteers (for example those that were never part of the corporate teams). I can see that authors have made a small disclaimer that it is hard to track changes in the corporate editing teams but I am sure that some illustrative examples could be found.

We are certain that illustrative examples can be found for any number of edit types, but as I often warn with OSM data: If you search hard enough, you can find an example to make any claim (a general criticism of big data science). These are fascinating questions but require a more complicated analysis infrastructure to answer at the global scale we are quantifying in this paper (we are currently building such an infrastructure).

These questions are proposed in our Future Work regarding local impact and map-seeding which would look to identify areas where corporate editors are active first and then other mappers maintain the data. At the moment, we think it is too early to draw definitive conclusions about these types of activities because corporate editing of this magnitude is so new. The coming years, however, will provide a rich dataset for this investigation.

Table 1 gives comparison of the sizes of teams and the number of their contributions. It does not say the years of active contributions and it does not compare the numbers with number of total OSM edits. The Figure 4 does that, but for the areas where companies are mapping. Given that almost the whole world is is covered by the edits it seams that corporate edits are overtaking the OSM. How does that compare to global number of edits/new objects/existing objects? That would be very informative!g, especially for a naive reader. Maybe even giving some trends to see if corporate editing is literally overtaking traditional voluntarism. That would also give some insights into voluntary contributions motivations eg. "Do mappers try to build up mapping experience so they can be employed like professional OSM mappers?"

You are right that we keep talking about this rise in Corporate editing, show the map, but do not show the other side. We have added both a “non-corporate” radar chart to Figure 4 (Figure 4k) and a table of the total number of edits performed by corporate-editors (table 2). In combination, the table describes how many edits are happening each year of each type, and then Figure 4k describes the relative impact on the map in terms of percentage of total features edited in these areas. We find this to be more informative and accurate representation than a global count of edits because it better accounts for where corporations are editing.

As for insights into voluntary contribution motivations, this is a fascinating question, but this is another question that cannot be accurately assessed from the editing record (database) alone. In the future, a survey of all corporate-editors would be fascinating. As it currently stands, most of these editors have a very generic description in their OSM user profiles, likely templated by the company:

“I am <name> from <place>. In my free time, I like to <hobby 1> and <hobby 2> and am happy to work on <company> mapping projects. Learn more about our mapping efforts at <link>.”

While motivations for mapping have been studied before (Budhathoki and Haythornthwaite 2013), the environment is clearly changing quickly and perhaps we are overdue for another far-reaching mapping motivation study. Effective, representative surveys of OSM contributors, however, is another complicated task in itself.

We have written a new discussion that comments on the idea of “building up mapping experience” to be hired as professional mappers: something that Budhathoki and Haythornthwaite even identified as early as 2013.

Even though very well written, without covering at least some of the previously mentioned topics and diving deeper into the data, the paper itself might be more appropriate for a conference than for a peer reviewed journal. Some of the new text could be used to replace the somewhat lengthy four pages introduction section. With some of the suggested improvements I would be happy to move this paper forward.Thank you for your review, we have modified the paper to address one of your questions and have explained here the nuances of the other research questions you proposed. We hope that future research and different methodologies will be able to more accurately tackle some of these more complex questions. We hope these new data and explanations are satisfactory to you to consider moving this paper forward.

I was able to spot only one tiny error on line 422 "...OSM might continue to function as the definitive instance of an open geospatial data base." Writing is excellent. Thanks for catching this, we reworded this sentence and took this phrase out entirely.

Reviewer 4 Report

This is an interesting article about the role of corporate editors on OpenStreetMap. The topic is relevant and timely, and the article provides new insight and relevant discussion. In some places, more specific examples and references are needed.

Comments

Abstract
Aside from identifying the rise of corporate editing, the abstract does not make a strong statement about the authors' original findings. Is there a statistic to support the claims about the rise of corporate editing? Can you highlight one or more main questions and concerns for the OSM community and researchers? The abstract could do a better job of highlighting the main take away points.

Introduction

I'm not sure if the graphs about participation inequality helped frame the paper. That topic is well-covered elsewhere. The material about the diversity of contributions, including for-profit interests, is very relevant.

Line 54 "the rapidly increasing number of paid-editors on the platform is new and has become a contentious issue for some in the community"
- I'm interested to see supporting evidence.

Line 60 - "In this article, we identify the corporations that transparently employ teams of professional editors, amounting to a total of ten"
- Awkward sentence structure.

Table 1. Is it possible to summarize in a few words a bit more about what types of activities these corporations are involved in as they relate to OSM?

Line 222
"One proxy for corporate interest and engagement in OSM is sponsorship of the OpenStreetMap Foundation (OSMF)."
Why is this a good metric? Why do companies who use OSM support OSMF and what if they don't?

Line 223
"esri" - is that the correct capitalization? On their website they seem to have a capital "E" most of the time.

"The aim of their Open Maps Team is to work closely with the OSM  community to improve data quality in places of strategic importance to Microsoft [52]. The team  coordinates their activities through GitHub where their team members and projects are listed. Microsoft’s commitment to OSM is an extension of their support of open source projects [50, 51]. "

These sentences seem very vague. Is there an example, or any other specifics, you can provide? 

Paragraph starting on line 272
This is a good level of detail. Nice to see a story like this.

Line 431. "However, another important argument comes from the  431
point of view of bias toward self-serving interests: are corporations introducing geographic bias into the map?"

Geographic and thematic bias. Shown by your Figure 4, and implied in the next sentence "As the map continues to be filled in, will corporate interests have too much voice in what and where gets mapped?" It would be great to see Figures 3 and 4 discussed or at least mentioned.

Author Response

Thank you for your review. We have addressed your comments point-by-point below and the new version of the document includes the mentioned changes with “track-changes” on so you can see the difference.

Aside from identifying the rise of corporate editing, the abstract does not make a strong statement about the authors' original findings. Is there a statistic to support the claims about the rise of corporate editing? Can you highlight one or more main questions and concerns for the OSM community and researchers? The abstract could do a better job of highlighting the main take away points.

We have added more tangible findings into the abstract and re-worded the takeaway to be more straightforward. The abstract is now at the 200 word limit.

I'm not sure if the graphs about participation inequality helped frame the paper. That topic is well-covered elsewhere. The material about the diversity of contributions, including for-profit interests, is very relevant.

We think that the participation inequality is important background context on the community that might not be as obvious to the average reader. We made an effort not to dwell on it or speculate, but rather mention it, cite other work, and show a figure to establish context. Another reviewer specifically mentioned this to be helpful.

Line 54 "the rapidly increasing number of paid-editors on the platform is new and has become a contentious issue for some in the community" - I'm interested to see supporting evidence.

While this is difficult to quantify, the tone of the mailing lists and the sentiment and behaviors of some community members at State of the Map Conferences is our supporting evidence.

Line 60 - "In this article, we identify the corporations that transparently employ teams of professional editors, amounting to a total of ten" - Awkward sentence structure

Agreed, we fixed it.

Table 1. Is it possible to summarize in a few words a bit more about what types of activities these corporations are involved in as they relate to OSM?

We touch on the involvement/relationship between OSM and these companies in Section 3.1.1. We have tried to avoid giving preferential treatment to any company by over-explaining them in the context and have instead provided all of the relevant links in footnotes and in Table 1.

Line 222 "One proxy for corporate interest and engagement in OSM is sponsorship of the OpenStreetMap Foundation (OSMF)." Why is this a good metric? Why do companies who use OSM support OSMF and what if they don't?
Reviewer 1 suggested to delete this introductory paragraph since it distracts from the 10 companies at hand. The questions proposed here also prompt further inquiry. We chose to delete this paragraph and focus on the known corporate editors and not speculate about OSMF engagement.

Line 223 "esri" - is that the correct capitalization? On their website they seem to have a capital "E" most of the time.

Thanks for pointing this out, this sentence is no longer in the paper.

"The aim of their Open Maps Team is to work closely with the OSM  community to improve data quality in places of strategic importance to Microsoft [52]. The team coordinates their activities through GitHub where their team members and projects are listed. Microsoft’s commitment to OSM is an extension of their support of open source projects [50, 51]. " These sentences seem very vague. Is there an example, or any other specifics, you can provide?

We have added another sentence better describing their (and other corporations) use of GitHub issues to track mapping projects and engage the community in discussions about the projects.

Paragraph starting on line 272: This is a good level of detail. Nice to see a story like this. Thank you, it is these types of details that highlight the contentious nature of corporate-involvement in OSM. These speculations by the community are inspired by (often well deserved, healthy) skepticism of corporate involvement.

Line 431. "However, another important argument comes from the point of view of bias toward self-serving interests: are corporations introducing geographic bias into the map?" Geographic and thematic bias. Shown by your Figure 4, and implied in the next sentence "As the map continues to be filled in, will corporate interests have too much voice in what and where gets mapped?" It would be great to see Figures 3 and 4 discussed or at least mentioned.

The discussion and conclusion have been re-ordered. In doing so, we have more tie-in to Figures 3 and 4 in this section.

Round 2

Reviewer 3 Report

Thank you for your comments and fast response.

Are there observable patterns for corporate editing except locations where they occur? For example, if Apple's contributions are observed, are all mapped roads done in the same fashion, using high or low estimated accuracy (many nodes or fewer nodes), always the same or different thematic attributes. I would expect that such corporate editing has some rules in order to make data more uniform so that more advanced algorithms can be used over mapped data. For example, when roads, mapped by Apple and Amazon are compared, who does more detailed job, even if the conclusion would come from couple of examples?

This is an interesting question with its own potential research agenda: company v. company. Particularly, could company-specific editing activities be meaningfully extracted, if so, could they be used to train ML models? Though it seems too speculative without insider-corporate knowledge whether or not these editors are mapping in a specific way to train algorithms down the road.

At this point, we do not think that there is enough known information about corporate-editing to start comparing the quality of one company’s work to another without more context. Since each company is focusing on different projects in different regions, this comparison does not seem rigorous. Though there are certainly great questions about impact and quality to be answered in this line of investigation, especially if we can find a specific job or task where corporations are naturally competing in OSM (and what would that mean?).

We have, however, added Table 2 that gives a better breakdown of the editing totals and growth per year by corporate editors.

I think that my comment has been misinterpreted and it was probably my fault. There is huge difference if an editor inserts only a way that represents for example highway=primary and does not add additional thematic data. I am speculating that companies' incentive for this type of mapping is motivated by desire to use such data within their services (not exclusively ML algorithms) so they want to make it more uniform. Therefore, I think that corporate edits follow in-house rules that would be visible across the edits. Otherwise they wouldn't be having OSM mapper on a payroll and would be just donating money to the community. Having the infrastructure to carry on this research, presented in the paper, I was thinking that it would be beneficial for the paper if you try to explore that part as well. It could be simple as counting average and median number of nodes per road (way) and which tags mostly occur. That would already produce some knowledge about a pattern. I find that such information would be very important in this topic. Otherwise, if we talk just about an edit, it can be as simple as moving one node compared to inserting a major highway.

Nevertheless, Table 2 makes huge difference and helps to better understand what companies focus on. Even though you have mentioned that focus on improving road infrastructure is common pattern in the process of map development, Table 2 data, presented in absolute numbers, is great contribution of this paper.

Majority of the corporate OSM mappers have had significant experience with OSM before they became members of such teams. Which company tries to hire the most of experienced editors. It would be an important information, are those the big companies like Amazon and Apple which are not primary OSM users or those are companies that base their business on OSM data like Mapbox and GraphHopper?

We agree, this is a very interesting question, but our methodology is not set up to answer this question (well).  While we can identify a number of mappers that were active in the community before through their username, this is only possible if they use the same username once they get hired. Companies like Facebook have specific usernames, such as VL0001 and VL0002 that employees use when editing. We cannot track if the individual behind this username was an active mapper before joining from the data alone.  Furthermore there are many other ways to be familiar with OSM data other than map editing that could make someone valuable to a company data-team. This question would best be answered with surveys and interviews with actual data-teams than speculated from the data. Per another reviewer’s suggestion, we have removed the speculation in this sentence in the paper and instead only report the known data. We hope future research will do these surveys.

Thank you for this comment, it satisfies me completely, though you might have mentioned in the paper the potential difference in usernames that you have found.

Even though locations of corporate edits are important part of the research, sections 3.2.1. (the second one, it should be 3.2.2.) and 3.2.3., and Figure 4 appear to be the core of the research. What is missing is the nature of the edits. Do they add tags and leave out geometry as is or only fix geometries? It doesn't have to be quantified data, giving the estimation of such edits would be very important. Also, are there examples where corporate edits are fixed by regular volunteers (for example those that were never part of the corporate teams). I can see that authors have made a small disclaimer that it is hard to track changes in the corporate editing teams but I am sure that some illustrative examples could be found.

We are certain that illustrative examples can be found for any number of edit types, but as I often warn with OSM data: If you search hard enough, you can find an example to make any claim (a general criticism of big data science). These are fascinating questions but require a more complicated analysis infrastructure to answer at the global scale we are quantifying in this paper (we are currently building such an infrastructure).

These questions are proposed in our Future Work regarding local impact and map-seeding which would look to identify areas where corporate editors are active first and then other mappers maintain the data. At the moment, we think it is too early to draw definitive conclusions about these types of activities because corporate editing of this magnitude is so new. The coming years, however, will provide a rich dataset for this investigation.

Thank you for this comment. I do not completely agree with it. For example, if you count 1000000 corporate edits but then 50% of those are overridden by non-corporate editors, and that fact is not detected in the research, then such results would be hiding a lot. That is why I have suggested that showing couple of examples would be illustrative if they are supported by proper counting of the same/similar cases. Of course that, if you find only one or couple of such examples in the whole dataset, it can then be considered as negligible and should not be included in the results. But given that research focuses on corporate edits as increasing corporate activity in the OSM community, mentioning this as a future research is appropriate.

Table 1 gives comparison of the sizes of teams and the number of their contributions. It does not say the years of active contributions and it does not compare the numbers with number of total OSM edits. The Figure 4 does that, but for the areas where companies are mapping. Given that almost the whole world is is covered by the edits it seams that corporate edits are overtaking the OSM. How does that compare to global number of edits/new objects/existing objects? That would be very informative!g, especially for a naive reader. Maybe even giving some trends to see if corporate editing is literally overtaking traditional voluntarism. That would also give some insights into voluntary contributions motivations eg. "Do mappers try to build up mapping experience so they can be employed like professional OSM mappers?"

You are right that we keep talking about this rise in Corporate editing, show the map, but do not show the other side. We have added both a “non-corporate” radar chart to Figure 4 (Figure 4k) and a table of the total number of edits performed by corporate-editors (table 2). In combination, the table describes how many edits are happening each year of each type, and then Figure 4k describes the relative impact on the map in terms of percentage of total features edited in these areas. We find this to be more informative and accurate representation than a global count of edits because it better accounts for where corporations are editing.

As for insights into voluntary contribution motivations, this is a fascinating question, but this is another question that cannot be accurately assessed from the editing record (database) alone. In the future, a survey of all corporate-editors would be fascinating. As it currently stands, most of these editors have a very generic description in their OSM user profiles, likely templated by the company:

“I am <name> from <place>. In my free time, I like to <hobby 1> and <hobby 2> and am happy to work on <company> mapping projects. Learn more about our mapping efforts at <link>.”

While motivations for mapping have been studied before (Budhathoki and Haythornthwaite 2013), the environment is clearly changing quickly and perhaps we are overdue for another far-reaching mapping motivation study. Effective, representative surveys of OSM contributors, however, is another complicated task in itself.

We have written a new discussion that comments on the idea of “building up mapping experience” to be hired as professional mappers: something that Budhathoki and Haythornthwaite even identified as early as 2013.

Thank you for this extensive comment. Figure 4k along with the first part of the Discussion section now shreds more light about the overall focus of the corporate editing. Even though the previous version was discussing the corporate edits main area of interest, addition to Figure 4 now quantifies this fact. The added discussion about the possible change of motivation of the mappers is excellent.

Even though very well written, without covering at least some of the previously mentioned topics and diving deeper into the data, the paper itself might be more appropriate for a conference than for a peer reviewed journal. Some of the new text could be used to replace the somewhat lengthy four pages introduction section. With some of the suggested improvements I would be happy to move this paper forward.Thank you for your review, we have modified the paper to address one of your questions and have explained here the nuances of the other research questions you proposed. We hope that future research and different methodologies will be able to more accurately tackle some of these more complex questions. We hope these new data and explanations are satisfactory to you to consider moving this paper forward.

I find that the paper, after the newly introduced changes, frames corporate edits much better into the OSM ecosystem. Numbers and comparisons that are given for the year 2018 actually give proof that the timing and significance of the paper is much higher than I initially thought. That will also influence my overall assessment of the paper. Therefore I am happy with the changes and comments and will accept this paper in the current form.